# See-N-Seq: RNA sequencing of target single cells identified by microscopy via micropatterning of hydrogel porosity

Jeong Hyun Lee[1,2], Emily S. Park [1,2], Jane Ru Choi [1,2], Kerryn Matthews[1,2], Alice V. Lam[1,2], Xiaoyan Deng[1,2], Simon P. Duffy[1,2,3] & Hongshen Ma [1,2,4,5 ✉]

Single cell RNA sequencing has the potential to elucidate transcriptional programs underlying key cellular phenotypes and behaviors. However, many cell phenotypes are incompatible with indiscriminate single cell sequencing because they are rare, transient, or can only be identified by imaging. Existing methods for isolating cells based on imaging for single cell sequencing are technically challenging, time-consuming, and prone to loss because of the need to physically transport single cells. Here, we developed See-N-Seq, a method to rapidly screen cells in microwell plates in order to isolate RNA from specific single cells without needing to physically extract each cell. Our approach involves encapsulating the cell sample in a micropatterned hydrogel with spatially varying porosity to selectively expose specific cells for targeted RNA extraction. Extracted RNA can then be captured, barcoded, reverse transcribed, amplified, and sequenced at high-depth. We used See-N-Seq to isolate and sequence RNA from cell-cell conjugates forming an immunological synapse between T-cells and antigen presenting cells. In the hours after synapsing, we found time-dependent bifurcation of single cell transcriptomic profiles towards Type 1 and Type 2 helper T-cells lineages. Our results demonstrate how See-N-Seq can be used to associate transcriptomic data with specific functions and behaviors in single cells.

[1] Department of Mechanical Engineering, University of British Columbia, Vancouver, BC, Canada. [2] Centre for Blood Research, University of British Columbia, Vancouver, BC, Canada. [3] British Columbia Institute of Technology, Vancouver, BC, Canada. [4] School of Biomedical Engineering, University of British Columbia, Vancouver, BC, Canada. [5] Vancouver Prostate Centre, Vancouver General Hospital, Vancouver, BC, Canada. ✉email: hongma@mech.ubc.ca

Recent advances in single cell RNA sequencing (scRNA-seq) have revealed remarkable transcriptomic heterogeneity between single cells, which have been attributed to differences of state[1], function[2], fate[3,4], and spatial positioning[5]. However, most current methods for scRNA-seq involve indiscriminate co-encapsulation of single cells with single beads inside droplets[6,7] or nanowells[8,9]. While these approaches can profile large numbers of single cells, contextual information associated with each cell is lost, and must be inferred *post hoc* from sequence data. To better understand how gene expression drives specific functions and behaviors in single cells, there is a need for methods to directly associate single cell transcriptomic data with visually observed cell phenotypes.

Several methods have been developed to sequence specific single cells identified by imaging, including micropipette aspiration[10,11], laser capture microdissection[12], magnetic micro-rafts[13], and optofluidic transport (e.g. Berkeley Lights)[14]. However, a major bottleneck in these processes is that single cells must be physically extracted from the imaging substrate and transported to a separate container, which is technically challenging, time-consuming, and prone to cell loss. The need to extract single cells can be eliminated by performing sequencing reactions directly inside intact cells using in situ sequencing techniques, such as FISSeq and pad-lock sequencing[15,16]. However, the limited optical space in each cell severely restricts read length and reads per cell[15–18], and the use of random priming reverse transcription is subject to bias[19]. These approaches also require substantial processing time and cannot take advantage of the ever-improving throughput of next generation sequencing.

To address the challenge of sequencing specific single cells identified by imaging, we developed See-N-Seq, a simple strategy for selective single cell mRNA capture without needing to physically extract each cell. We achieve this goal by first dividing a cell sample among microwells in an imaging well plate. In each microwell, a photo-polymerized hydrogel encapsulates all cells except for a single target cell. Next, we capture mRNA from each target cell using oligo-dT beads, which are then extracted for downstream barcoding, reverse transcription, amplification, and library preparation. To demonstrate the potential of using See-N-Seq to sequence RNA from cells exhibiting complex visual phenotypes, we used this approach to sequence transcriptomes from immunological synapses formed by the conjugation of T cells and antigen presenting cells. Using model cells, we selected synapses based on CD3 localization and sequenced transcriptomes at 0, 4, and 24 h time points, which showed a time-dependent bifurcation towards Th1 or Th2 lineages. Our results demonstrate how See-N-Seq can be used to associate gene expression with specific visually observed phenotypes to reveal the transcriptional programs driving specific cell functions and behaviors.

## Results

See-N-Seq enables targeted transcriptome capture from single cells identified in imaging microwell plates. This process involves (1) dividing a cell sample among microwells of a standard 384-well imaging plate, (2) encapsulating cells in a porous hydrogel thin-film to fix their position, (3) imaging the cell sample to identify one target cell from each microwell, (4) embedding all non-target cells in second non-porous hydrogel polymerized by laser micropatterning, and (5) lysing target cells to extract mRNA using oligo-dT beads for subsequent barcoding, reverse transcription, PCR amplification, and sequencing library preparation (Fig. 1).

**Cell encapsulation in porous hydrogel thin-film**. Encapsulating the cell sample in a porous hydrogel thin-film is required to fix

cell position in the imaging microwell plate. This step is essential because cells can potentially move between the imaging and laser micropatterning steps due to motility and fluid flow. Anchoring the cell sample to the imaging surface is also necessary because the prepolymer for the non-porous hydrogel has greater density than cells, which could dislodge the cells by buoyancy unless they are attached to the imaging surface.

The porous hydrogel is made using a high-molecular weight PEGDA polymer mixture originally developed for lossless imaging[20]. This pre-polymer has lower density than cells, and thereby allow cells to be sedimented to the bottom of imaging well plates by centrifugation. The pre-polymer is crosslinked using 375 nm UV to form a 100–150 μm thick porous hydrogel thin-film that could be readily penetrated by large biomolecules, such as nucleic acids and proteins. We previously showed this hydrogel thin-film can be used to retain cells without loss during antibody staining and washing steps in immunofluorescence protocols. Other surface anchoring methods, such as chemical adhesion using poly-L-lysine, could also be used, but is less desirable because these methods are incompatible with some cell types and can affect cell morphology, behavior, and gene expression[21]. In contrast, encapsulating cells using a porous hydrogel thin-film preserves cell morphology and is compatible with all cell samples. Transcriptional changes resulting from exposure to these polymers are also limited by minimizing exposure time (<3 min) before ethanol fixation.

**Patterning non-porous hydrogel inside the porous hydrogel for single cell isolation**. To selectively isolate RNA from a single target cell in each microwell, all non-target cells are embedded in a micropatterned non-porous hydrogel in such a way to expose a single target cell. This hydrogel is formed by photopolymerization of low-molecular weight PEGDA prepolymer (MW = 250 Da). This prepolymer is solubilized in ethanol and readily penetrates into the porous hydrogel thin-film. Micropatterning of the non-porous hydrogel is performed using a UV laser scanner integrated into an inverted fluorescence microscope, which is also used for imaging. The laser scanner comprises of a 405 nm diode laser coupled via a ferrule core optical fiber and collimator to a 2D galvo-galvo scanner. The galvo-galvo scanner angles the laser beam, which is converted to a lateral offset using a telecentric scan lens. The offset laser beam enters the microscope via a scan lens and is focused on the sample by the microscope objective. Computer control of the laser position is accomplished using custom software to control the laser intensity, galvo-galvo scanner angle, and the microscope stage. Patterned laser stimulation in each microwell is performed by first generating a bitmap image of the desired pattern, which is then raster scanned using the galvo-galvo scanner.

To establish the spatial resolution of micropatterning non-porous hydrogels inside porous hydrogels, we encapsulated a cell sample in the porous hydrogel thin-film, and then micropatterned the non-porous hydrogel by raster scanning a circular void pattern. A 10× objective can be used to form 50 μm voids, while further patterning using a 20× objective can reduce the void size to 12 μm, which is sufficient for isolating single cells in suspension (Fig. 2a–c). The laser spot size formed using a 20× objective is ~1 μm in diameter, but the patterning resolution is limited to ~12 μm because of the spatial propagation polymerization from the nucleating laser spot. To confirm the porosity of the patterned hydrogel microstructure, we added 10 kDa FITC-dextran to the well, which was able to penetrate the porous circular void, but not penetrate the surrounding non-porous hydrogel (Fig. 2d, e).

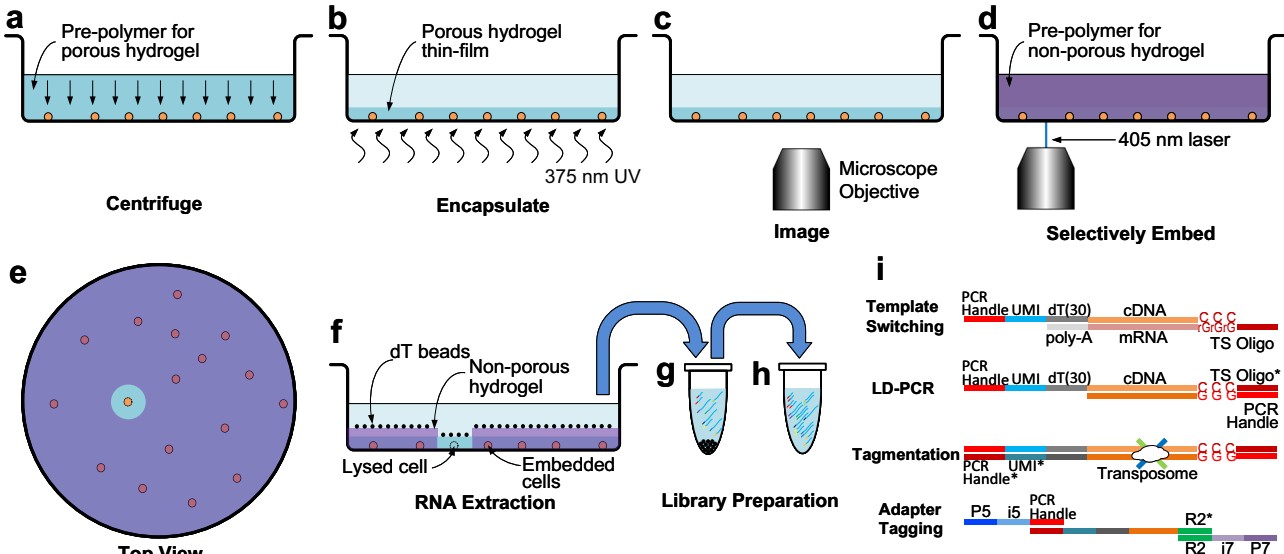

**Fig. 1 Selective single cell RNA sequencing using See-N-Seq. a** The cell sample is distributed in imaging microwell plates with the pre-polymer for the porous hydrogel and aligned by centrifuging. **b** The cells are encapsulated in a porous hydrogel thin-film formed by exposure to 375 nm UV light. **c** Encapsulated cells are imaged by microscopy to identify a target cell in each well. **d** The pre-polymer for the non-porous hydrogel penetrates inside the porous hydrogel and is micropatterned using a scanning 405 nm laser. **e** (Top view) All cells except for the target cell are embedded in the non-porous hydrogel. **f** mRNA from the target cell is captured by oligo-dT beads. **g** The oligo-dT beads are transferred to a tube where mRNA is eluted, UMI-barcoded, and reverse transcribed with a template switching oligo. **h** The cDNA library is amplified by long distance PCR (LD-PCR). **i** Library preparation steps include template switching, LD-PCR, tagmentation, and adaptor tagging.

**Screening and processing throughput**. The screening throughput of See-N-Seq derives from the cell seeding density in standard 384-well plates. Based on the laser patterning resolution of 12 μm, a cell seeding density of ~2000 cells per well provides sufficient space between cells for RNA isolation from single cells. The processing throughput to extract single cell RNA from target cells is determined by the time required for microscopy and polymerization. For the configuration described here, acquiring 4-channel images from each well in a standard 384-well plate requires 8 s at 10× magnification or 10 s at 20×. Polymerizing the non-porous hydrogel requires 15 s per well. Therefore, a full 384-well plate can be processed in ~2.5 h in order to select 384 single cells by screening ~750,000 cells using 10X magnification imaging. A potential bottleneck in this analysis is the cell selection process, which may require higher magnification or greater imaging time for complex phenotypes.

**Selective nucleic acid extraction from target single cells**. To validate targeted cell lysis and RNA extraction, we encapsulated a mixture of mCherry- and EGFP-expressing UM-UC13 cells in the porous hydrogel. We then micropatterned the non-porous hydrogel to selectively expose a single target cell. Adding lysis buffer eliminated fluorescence of the target cell, while all other cells retained their fluorescence, although, fluorescence intensities of non-target cells were reduced due to dehydration (Fig. 2f–h). We then selectively extracted RNA from individual mCherry- and EGFP-expressing UM-UC13 cells for analysis using qPCR. As expected, mCherry single cells showed detectable mCherry signal with an absence of EGFP signal, and vice versa for EGFP single cells (Fig. 2i).

**Selective RNA sequencing of target single cells**. To perform selective single cell transcriptome sequencing, we isolated one single cell from each microwell using hydrogel micropatterning as before. After adding the lysis buffer, the released mRNA is captured using oligo-dT beads (Fig. 1f), which are transferred to PCR tubes where the oligo-dT beads are eluted (Fig. 1g, h). The mRNA is reverse transcribed using oligo-dT primers containing a PCR handle, a 16-bp unique molecular index (UMI), as well as a template switching oligo (TSO) to attach a PCR handle at the 5'end of the cDNA (Fig. 1i). The resulting cDNA library is then amplified via long-distance PCR (LD-PCR) (Fig. 1h). The amplicons are cleaned up, tagmented, and tagged with sequencing adaptors. The cDNA library from indexed multiple single cells could then be pooled and sequenced, enabling direct tracing from image of targeted cells to sequence data (Fig. 1i, Supplementary Fig. S1). We prepared sequencing libraries for both Jurkat and Raji single cells (Jsc, Rsc), as well as bulk cell samples (~1000 cells; Jb, Rb). Each sample was sequenced using 3.56 ± 0.96 million reads. After analyzing the sequencing data, we found an average of 0.93 ± 0.47 million unique transcripts from bulk samples and 0.39 ± 0.11 million unique transcripts from single cell samples. The detected number of transcripts from single cells is similar to estimates of the total number of transcripts expressed by these cells[22].

To confirm that target single cell transcriptome sequencing can be performed without contamination, we used See-N-Seq to isolate single cell RNA from 1:1 mixtures of UM-UC13-EGFP human cancer cells and NIH-3T3 mouse fibroblasts (pre-labeled using Vybrant DiL orange). We generated RNA sequencing libraries from five human and five mouse cells. Each cell was sequenced using 1.73 ± 0.82 million reads. Following sequencing, all reads were aligned against reference assemblies for both human (hg19) and mouse (mm10) genomes to obtain read counts for each annotated transcript. Sequencing libraries from human cells had a mean UMI count of 0.13 ± 0.06 million with a purity range of 94.4–99.4%, while sequencing libraries from mouse cells had a mean UMI count of 0.10 ± 0.08 million with a purity range of 93.5–99.1%. These purity levels are in line with other single cell sequencing methods[6,8] (Fig. 2j). These results confirm that See-N-Seq is able to extract RNA from target single cells without contamination from non-target cells. We further used mean UMI count to assess RNA yield from difference cell

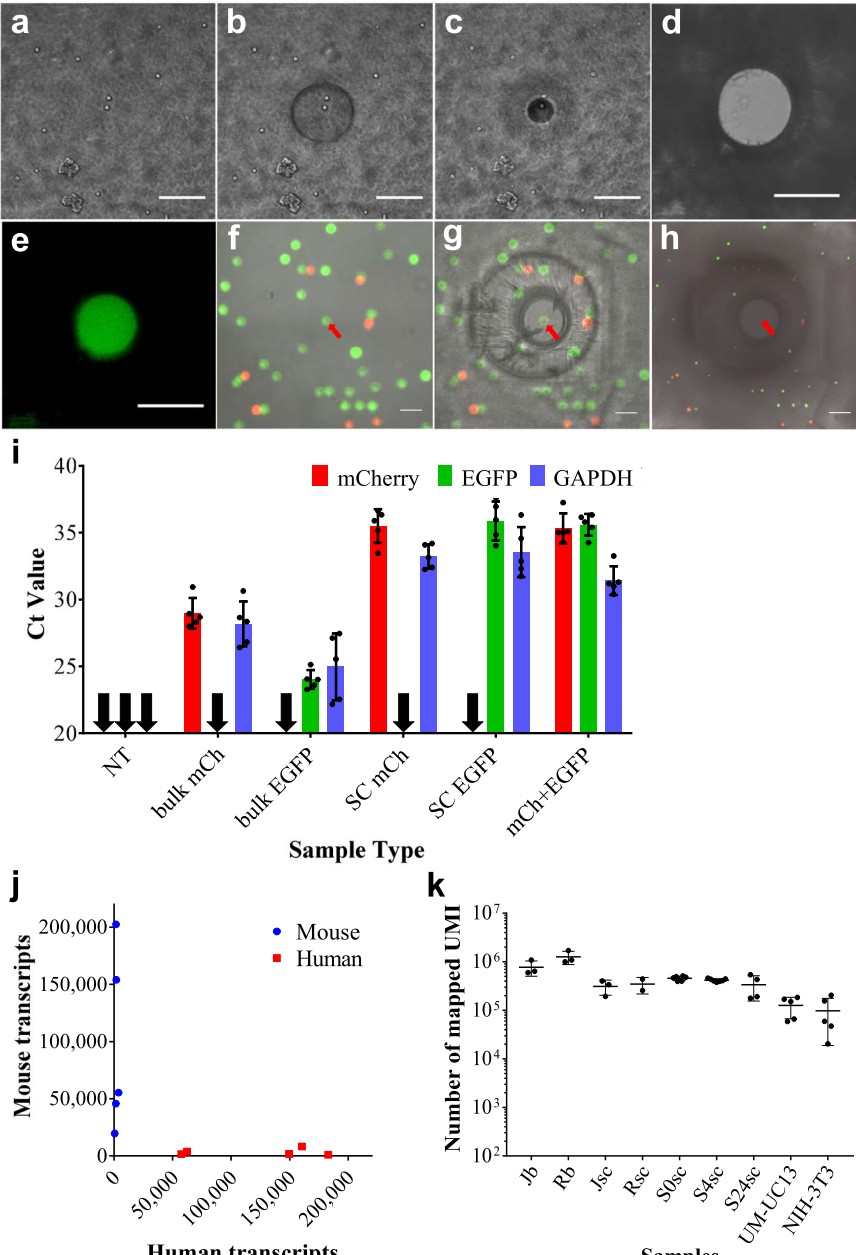

**Fig. 2 Micropatterning hydrogel porosity to enable selective nucleic acid extraction. a** Cells encapsulated in a porous hydrogel. **b** Micropatterning a 50 μm void in the non-porous hydrogel using a 10× objective. **c** Further micropatterning a 12 μm void using a 20× objective. **d** Bright-field micrographs of a 50 μm disk of porous hydrogel surrounded by non-porous hydrogel. **e** Diffusion of FITC-Dextran (Mw 10 kDa) into the porous hydrogel disk. **g, h** Selective single cell lysis using UM-UC13-mCherry (mCh) and UM-UC13-EGFP (EGFP) cells shown using overlaid bright-field and fluorescence micrographs. **f** mCh and EGFP cells encapsulated in porous hydrogel. **g** All non-target cells are embedded in non-porous hydrogel, while exposing a single target EGFP cell. **h** After addition of cell lysis buffer, only the target EGFP cell was lysed. Fluorescence of non-target cells is reduced due to dehydration, but not eliminated. **i** qPCR for RNA selectively extracted from mixed mCh and EGFP cell samples including no-template control (NT), bulk mCh, bulk EGFP, single cell mCh, single cell EGFP, as well as one mCh and one EGFP cell. Black arrows indicate target cDNA undetectable. Ct values are averaged from $N = 5$. **j** See-N-Seq analysis from mixtures of mouse and human cells. The scatter plot shows the number of unique human and mouse transcripts associating to each sample. Blue dots indicating mouse single cells while red dots indicating human single cells. $N = 5$. **k** Mapped UMI counts per each sample group. All scale bars = 50 μm.

types, including single suspension (Jurkat and Raji), pairs of suspension cells (Jurkat-Raji cell pairs), as well as enzymatically disassociated adhesion cells (NIH-3T3 and UM-UC13). Our data show relatively consistent UMI counts within each cell type, which suggest that our process is extracting a relatively consistent fraction of the total mRNA from each cell (Fig. 2k).

**Using See-N-Seq to sequence complex phenotypes and transient processes.** To demonstrate the ability to sequence transcriptomes from complex image-defined phenotypes, we used See-N-Seq to isolate mRNA from cell pairs engaged in immunological synapses. An immunological synapse is a physical junction between helper T cells (Th) and antigen-presenting cells (APCs) where the T cell receptor (TCR/CD3) localizes to the cell-

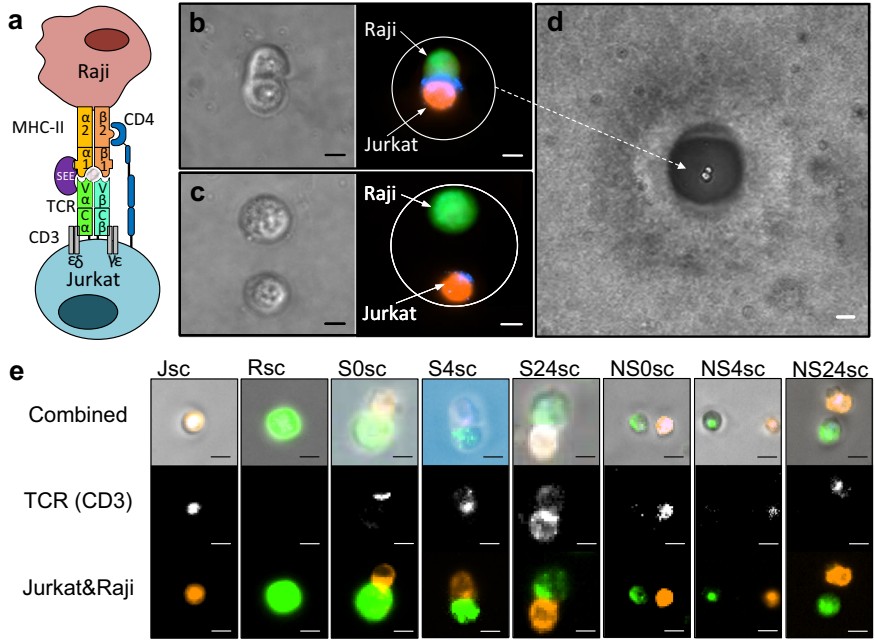

**Fig. 3 Using See-N-Seq to isolate complex cell phenotypes for RNA sequencing. a** Schematic of an immunological synapse formed between Jurkat and Raji cell using *Staphylococcus aureus* enterotoxin E (SEE) superantigen. **b** Micrographs of an immunological synapse formed between Jurkat cell (orange) and SEE treated-Raji cells (green). TCR is identified using anti-CD3 (blue) localized at the interface of the two cells. Scale bar = 5 μm. **c** Micrographs of a non-synapsing Jurkat-Raji cell pair. Scale bar = 5 μm. **d** Jurkat-Raji cell pair in B isolated via See-N-Seq. Scale bar = 25 μm. **e** Micrographs of sequenced cells or cell pairs including single Jurkat (Jsc) and Raji (Rsc) cells; synapsing cells at 0, 4 and 24 h (S0sc, S4sc, S24sc); as well as non-synapsing cells at 0, 4 and 24 h (NS0sc, NS4sc, NS24sc). Scale bar = 5 μm.

cell interface to confirm antigen recognition, and then initiate T cell activation and differentiation based on signaling from co-stimulatory (e.g. CD28) and co-inhibitory (e.g. CTLA4, PD-1) receptors[23–25]. Following the formation of immunological synapses, T cells may differentiate into multiple subtypes (e.g. Th1 and Th2) that coordinate distinct cell mediated or humoral adaptive immune responses. T cell activation has largely been elucidated by scRNA-seq following activation with anti-CD3/CD28 beads[26]. However, previous studies have not accounted for co-regulatory receptor signaling present in a natural immunological synapse[10,27,28]. Elucidating transcriptional programs arising from the formation of immunological synapse has been a difficult challenge because of the need to verify synapse formation via CD3 localization at the cell-cell junction.

We induced the formation of antigen-independent synapses between Raji (antigen presenting cell) and Jurkat cells (T cell) using enterotoxin E (SEE) super-antigen (Fig. 3a). To determine the frequency of immunological synapses in this admixture, we performed flow sorting of heterotypic cell pairs. Raji cells are labeled using a membrane stain (Vybrant DiO green), while Jurkat cells are labeled using a membrane stain (Vybrant DiL orange) and a CD3 antibody. After gating for cell pairs, we found that ~1% of the flow cytometry events consisted of Jurkat-Raji cell pairs. We then imaged the heterotypic cell pairs isolated by flow sorting and we found CD3 localization in ~9% of these cell pairs, or 0.09% of total (Supplementary Fig. S2). Natural immunological synapses requiring antigen-specificity are likely to be found at an even lower frequency, and therefore require significant screening throughput to detect.

We used See-N-Seq to isolate transcriptomes from Jurkat-Raji immunological synapses by first encapsulating the cell mixture distributed across different microwells using the porous hydrogel. We then imaged the cell sample to identify specific phenotypes of interest, including Jurkat-Raji immunological synapse with CD3 localization at the cell-cell junction, as well as non-synapsing

Jurkat and Raji cells that were in close proximity but were spatially separated (Fig. 3b, c). We then patterned the non-porous hydrogel in order to isolate RNA from each phenotype. Specifically, we isolated RNA from synapsing Jurkat and Raji cell pairs formed after 0, 4, and 24 h of incubation (S0sc, S4sc, S24sc in Fig. 3e). As a control, we isolated RNA from single Jurkat cells, single Raji cells, as well as non-synapsing Jurkat and Raji cell pairs at 0, 4, and 24 h of incubation (Jsc, Rsc, N0sc, N4sc, N24sc, in Fig. 3e).

Analyzing the transcriptomic data, we found that transcriptomes of Jurkat and Raji single cells showed more variation than bulk samples (Supplementary Fig. S3a, b). Differential gene expression (DGE) analysis comparing single Jurkat and single Raji cells, presented as mean-divergence (MD) plots, showed 424 upregulated and 890 downregulated genes (Fig. 4a; *p* < 0.05, FDR corrected). The most significant upregulated genes in Jurkats included those involved in T cell signal transduction and cytokine signaling (*CD3D, IFITM1, IL32*), T cell migration (*FLNC, CAPG*), as well as resistance to apoptosis and cell stress (*GSTP1, GPX1*).

The gene expression data for non-synapsing cell pairs (N0sc) showed a high degree of similarity when compared to concatenated transcriptome data from single Jurkat and single Raji cells (N0sc vs. Jsc + Rsc, Fig. 4b). This level of similarity was also observed when comparing non-synapsing cell pairs and synapsing cell pairs at 0 h (N0sc vs. S0sc, Fig. 4c), as well as between randomly selected synapsing cell pairs at 4 h (Fig. 4d and Supplementary Fig. S3f). At 4 h post synapse, we begin to see an appreciable amount of differential gene expression with 85 upregulated and 153 downregulated genes compared to the synapse gene expression profile at 0 h (S4sc vs. S0sc, Fig. 4e and Supplementary Fig. S3d, e). Key differentially expressed genes associated with intracellular signal transduction (*CCL17, ADGRG1, ROR1, CCL21, GNA13*), cell-cell junction (*FBF1, LIMS2, STRN, GJB2*), and synaptic transmission (*SYNJ2BP, CHRNB1, P2RX1, UNC13B, APBA3*) were highly enriched.

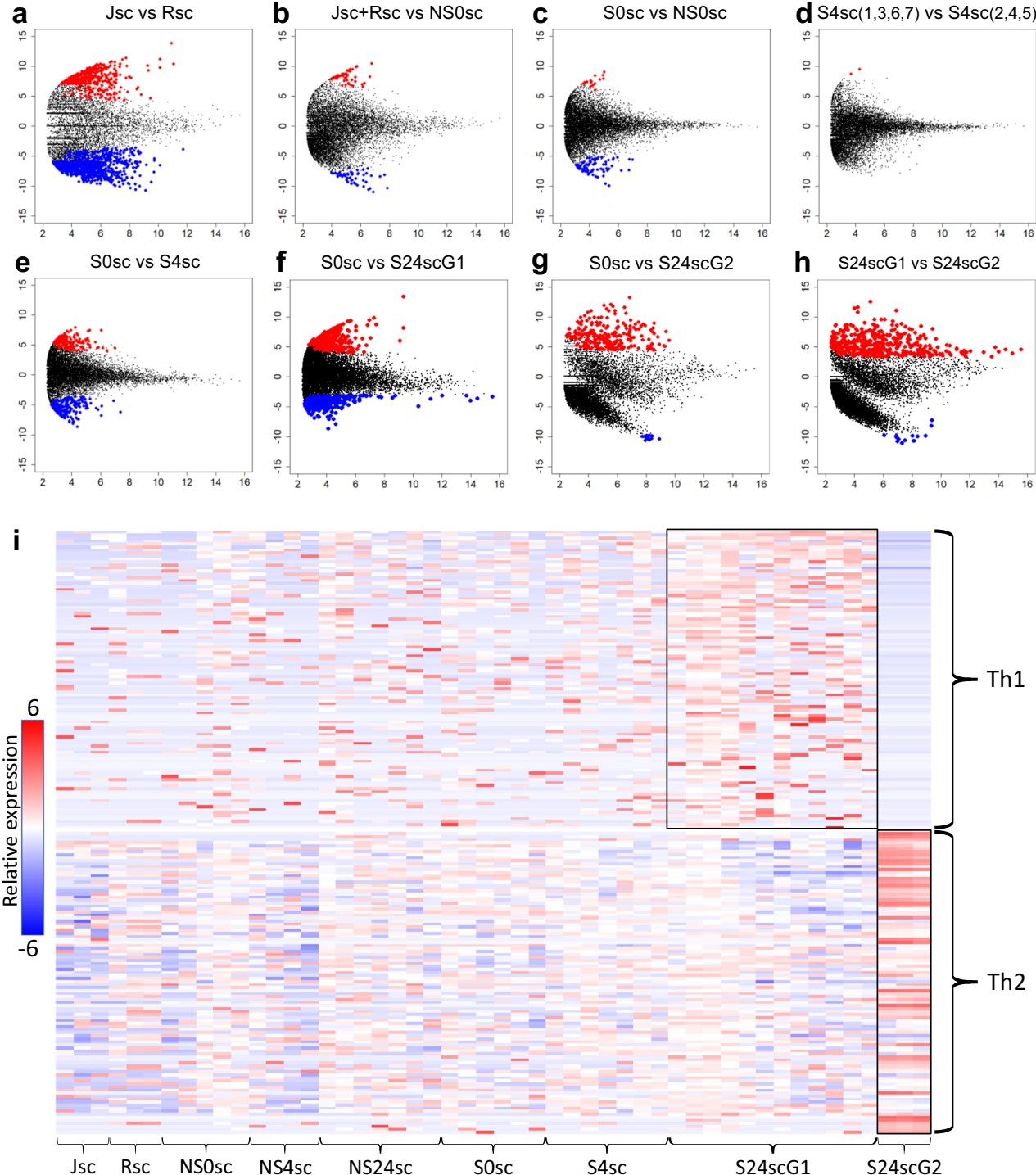

**Fig. 4 Differential gene expression data on Jurkat-Raji immunological synapse cell pairs and controls. a–h** Mean-Divergence plots comparing transcriptomes from Jurkat single cells (Jsc); Raji single cells (Rsc); synapsing cell pairs at 0, 4, and 24 h (S0sc, S4sc, S24sc); non-synapsing cell pairs at 0, 4, and 24 h (NS0sc, NS4sc, NS24sc). X-axis shows average log-CPM (counts per million). Y-axis shows log-fold-change between tested groups. Highlights indicate upregulated (red) and downregulated (blue) genes with $p < 0.05$. **a** Jsc vs. Rsc. **b** Concatenated Jsc and Rsc transcriptomes vs. NS0sc. **c** S0sc vs. NS0sc. **d** Comparing two randomly selected groups of S4sc. **e** S0sc vs. S4sc. **f** S0sc vs. synapse at 24 h group 1 (S24scG1). **g** S0sc vs. synapse at 24 h group 2 (S24scG2). **h** S24scG1 vs. S24scG2. **i** Heatmap showing log-CPM of putative genes for T helper cell type 1 (Th1) and type 2 (Th2).

At 24 h post-synapse, we found dramatically different gene expression profiles with 465 upregulated and 551 downregulated genes compared to the synapse gene expression profile at 0 h (S24sc vs. S0sc, Fig. 4f, g and Supplementary Fig. S3g, h). To compare gene expression between cells at 24 h, we fitted a linear model to the expression of genes by each cell pair and we calculated the Pearson

coefficient for each (Supplementary Fig. S4a). These data were used to generate a dendrogram using Unweighted Pair Group Method with Arithmetic mean (UPGMA, Supplementary Fig. S4b). Interestingly, cell clustering based on gene expression at the 24-h timepoint produced two distinct cluster groups, denoted S24scG1 and S24scG2. Cells clustered within S24scG1 exhibited universal

and exclusive over-expression of TBX21, which is the master-regulator of Th1 differentiation[29]. Conversely, cells within S24scG2 exhibited universal and exclusive over-expression of GATA3, a master regulator of Th2 differentiation[30]. These data were used collectively to infer that S24scG1 and S24scG2 constituted cells that differentially polarized to Th1 and Th2 subtypes. Further support for lineage-specific gene expression was obtained by identifying additional transcripts that have previously been reported as Th1 and Th2 transcriptional markers (Fig. 4i and Supplementary Table S1). In addition to TBX21, overexpression of GZMB, XCL1, XCL2 and TCIRG1/TIRC7 is particularly indicative of Th1 polarization of S24scG1 cells[31,32]. Overexpression of PRKCQ and Th2-specific proliferation factor are similarly particularly indicative of Th2 polarization in S24scG1 cells[33]. The capacity for Jurkat cells to differentiate toward a Th2 lineage under certain conditions has been previously reported[24]. However, the findings reported here are remarkable because they suggest that cells forming an immunological synapse in response to the same activating conditions can adopt divergent transcriptional programs after synapsing.

## Discussion

See-N-Seq addresses a key challenge in single cell transcriptomics by associating transcriptome data directly with observed single cell phenotypes. This capability is important for characterizing phenotypes that are rare or transient in nature, and cannot be inferred from transcriptomic data obtained by indiscriminate single cell sequencing[6,7] and flow sorting[34–36]. The process for See-N-Seq involves distributing cells into 384 well imaging plates where they can be rapidly screened using conventional high throughput microscopy. Single cell isolation is accomplished by selectively embedding non-target cells in non-porous hydrogel, using a laser micropatterning process. Isolation of single cells can be accomplished in 15 s and does not pose a risk of cell loss that may result from physical extraction. RNA from exposed target cells can be extracted using capture beads and reverse transcribed with UMI barcoding, which enables deep and unbiased transcriptome sequencing.

The selectivity of the current See-N-Seq process is limited by the resolution of the laser patterning process. The scanning laser system provides a focused spot size of ~1 μm, but the minimum patterning resolution is ~12 μm due to polymerization propagation. The patterning resolution limits the maximum cell density to ~2000 cells per well in a standard 3.2 ×3.2 mm well in a 384 well plate, which provides a screening throughput of 750,000 cells per well plate. The current patterning resolution also prevents the application of See-N-Seq for RNA extraction from individual adherent cells, which typically grow adjacent to each other on microwell plate substrates. Nonetheless, we showed that enzymatically disassociated adherent cells could still be sequenced using this approach (Fig. 2k). Extending See-N-Seq to adherent cells will likely require a combination of improvements in photopolymer chemistry and the optical system to improve patterning resolution.

Comparing to indiscriminate scRNA-seq, See-N-Seq represents a fundamentally different approach that enables hypothesis-driven characterization of rare or transient cell phenotypes, such as T cells engaged in an immunological synapse. Identifying rare or transient cell phenotypes from indiscriminate scRNA-seq data require knowledge of specific marker genes or using clustering algorithms to identify specific cell groups. However, marker genes are often not known or expressed consistently enough to identify key specific phenotypes from single cell transcriptomic data, while rare and transient cell types may not be sufficiently represented to be effectively clustered from the overall population. Rather than screening transcriptome datasets, See-N-Seq

provides a method to identify specific phenotypes from imaging. The screening throughput of this process is >750,000 cells from in each 384-well plate and this process can be completed in ~2.5 h.

Previous methods for selective single cells sequencing invariably rely on physically transferring single cells from one container to another. This approach is limited by a trade-off between screening throughput and processing throughput, which depend on the density at which the cells are plated on the imaging substrate. Methods like micropipette aspiration[10,11] and laser capture microdissection[12] permits higher screening throughput, but are extremely laborious and prone to cell loss, which dramatically limits sequencing throughput. Conversely, methods like microrafts[13] and Berkeley Lights[14], simplify the cell extraction process by plating cells at lower density, but dramatically limits screening throughput. By eliminating the need to physically extract single cells, See-N-Seq can provide both high screening throughput along with reasonable processing throughput.

The ability to associate transcriptomic data with observed phenotypes has the potential to enable new biological discoveries by elucidating of the transcriptional programs that drive specific cellular processes. One of the most interesting consequences of this capability is the potential to investigate information processing and decision-making at the single cell level with precisely controlled stimuli. As an example, we investigated the potential to sequence transcriptomes arising from the formation of T cell immunological synapses, which are particularly elusive because of the transient nature of these cell-cell interactions. T cell activation has been previously characterized using anti-CD3 and anti-CD28 bead activation[37]. However, this approach fails to integrate co-receptor signaling from antigen presenting cells that inform T cell decision-making to ultimately determine the type and strength of the cellular immune response[38–40]. Using model T cells and antigen presenting cells, we found gene expression profiles consistent with both T cell activation and differentiation following synapse formation. Interestingly, we observed bifurcated transcriptional states at the 24-h time point where T cell transcriptional profiles diverged towards either a Th1 or Th2 lineage. Future work can leverage See-N-Seq to study immunological synapses in human lymphocytes ex vivo to investigate how specific cell signaling stimuli contribute to cell fate determination in a variety of immunological processes. These insights would be particularly valuable in the context of T cell therapies where cell fate determination plays an important role in establishing favorable differentiation states and in vivo persistence[41].

See-N-Seq can also be applied more broadly to characterize cellular events that are inaccessible to current sequencing methods, owing to the simplicity of this method and the ability to directly extract mRNA from 384-well imaging plates. This approach can also leverage recent advances in this field to combine gene expression profiling with DNA analysis following whole genome amplification[42], analysis of methylation and chromatin accessibility (scNMT-seq)[43] or protein abundance (CITE-seq)[44] to enable multi-omics analysis of single cells. The use of standard microwell plates makes See-N-Seq scalable through integration with existing workflows for high throughput microscopy and robotic liquid handling.

Together, See-N-Seq provides a new dimension to single cell transcriptomics, where cells can be selected based on rare and transient cellular phenotypes that are then directly associated with transcriptome data in order to elucidate the transcriptional programs driving specific cell phenotypes and behaviors.

## Methods

**Cell lines**. Fluorescent human male bladder cancer cell lines, UM-UC13-EGFP and UM-UC13-mCherry, were a kind gift from Dr. Peter Black (University of British Columbia, Vancouver, Canada). These cells were maintained in Minimum

Essential Medium (MEM) culture media supplemented with 10% (v/v) fetal bovine serum (FBS) and 1% (v/v) penicillin-streptomycin in a humidified incubator at 37 °C and 5% $CO_2$. These cell lines were authenticated for transgene overexpression by immunofluorescence and PCR. Cells were harvested using 0.25% trypsin-EDTA, centrifuged at 200 x g for 5 min, and resuspended in phosphate buffered saline (PBS) to generate serial dilutions for the single-cell extraction experiments. Jurkat E6-1 male T cells (ATCC TIB-152) and Raji male B cells (ATCC CCL-86) were STR-authenticated cell lines obtained from ATCC. These cells were cultured at low passage and maintained in RPMI-1640 with 10% (v/v) FBS and 1% (v/v) penicillin-streptomycin. To generate immunological synapses, $2 \times 10^5$ Raji B cells were pre-incubated for 1.5 h with 2 µg/ml of purified superantigen *Staphylococcal aureus* enterotoxin E (SEE, Toxin Technology, Cat# ET404) and washed twice with PBS supplemented with 1% bovine serum albumin (BSA) and centrifugation at $200 \times g$ for 5 min. Jurkat cells were labeled with Vybrant DiO (Peak emission − 501 nm) and Raji with Vybrant DiI (Peak emission − 565 nm), according to the manufacturer's instructions. In addition, mouse anti-human CD3 (UCHT1) conjugated with Brilliant Violet-421 (BV421, blue) was added to T cells to visualize the TCR localization during immunological synapse. Stained cells were mixed and incubated in 40 µL culture media at 37 °C for the duration of each experiment.

**Hydrogel preparation.** Porous hydrogel pre-polymer was obtained from Milli-poreSigma (LMR001). Non-porous hydrogel was produced by combining 2% (w/v) MEHQ with 100% Poly(ethylene glycol) diacrylate (Mn 250) (PEG250-DA), and adding Irgacure 819 (Phenylbis(2,4,6-trimethylbenzoyl)phosphine oxide) to 1% (w/v). Each solution was freshly prepared prior to each experiment.

**Laser micropatterning system.** The custom laser micropatterning system consists of a 405 nm diode laser (50 mW, OBIS LX FP, Coherent), fiberport collimator (PAF2-A4A, Thorlabs), 2-axis galvo-galvo scanner (LSKGG4, Thorlabs), scan lens (CLS-SL, Thorlabs), and tube lens (TTL200, Thorlabs). Each component was assembled through a 60 mm cage and SM2 tube mounted at the rear port of a Nikon Ti-E microscope. The input laser is reflected by a dichroic mirror at the filter turret to the objective lens. The laser and galvo-galvo were controlled using NI-DAQ (PCIe-6363, NI) via custom software. Conventional photo-activation systems that provide region-of-interest scans using a 405 nm laser can also be used to to perform the micropatterning work described here.

**Diffusion test.** To assess the diffusion of 10 kDa FITC-dextran molecules through the porous hydrogel, a 50 µm-void pattern was formed using the non-porous inside the porous hydrogels. Briefly, the porous hydrogel pre-polymer was photo-polymerized and washed with 100% ethanol. Then, non-porous prepolymer was introduced and a 50 µm void pattern was created using laser micropatterning. The well was washed with 100% ethanol and PBS, and then 5% of FITC-dextran was added and incubated for 10 min. After incubation the excess FITC-dextran was removed and washed with PBS once. The diffusion image was captured using a Nikon Ti-E microscope.

**Cell encapsulation and laser micropatterning.** To encapsulate the cells in the porous hydrogel, the 40 µL cell suspensions in culture media were added into a 384-well glass bottomed imaging plate with 6.5 µL of the porous hydrogel pre-polymer (LMR001 MilliporeSigma). The 384 well-plate was centrifuged at $3300 \times g$ for 3 min (Accuspin 1 R, Fisher scientific), followed by exposure to 375 nm UV LED (Thor labs, Cat# M375L3) for 5 s. After photo-polymerization, non-polymerized hydrogel prepolymer was removed and washed three times with 50 µL of PBS. Cells encapsulated in porous hydrogel were imaged using a Nikon Ti-E fluorescent microscope to identify the location of specific target cells. Each well was washed ten times with 50 µL of 100% ethanol. At this step, cells are rapidly fixed by ethanol. Next, 40 µL of non-porous hydrogel pre-polymer was added to each well and the plate was centrifuged at $100 \times g$ for 5 min, allowing the prepolymer to penetrate the first porous hydrogel. Based on the previously stored location of the target cells, the 405 nm laser is scanned at pixel dwell time of 1~5 µs with laser power of 0.5~2.5 mW to polymerize all regions except for a void around the target cell. After photo-polymerization, the remaining non-porous polymer is removed and washed with 100% ethanol.

**RNA extraction, reverse transcription (RT) and qPCR.** After laser-micro-patterning, target cells were lysed (LiDS lysis buffer) and mRNA was extracted using the Dynabeads mRNA direct purification kit (Invitrogen), according to the manufacturer's protocol. Briefly, 20 µL LiDS lysis buffer was incubated in each well and incubated at room temperature for 5–10 s. Then, 10 µL of Oligo dT beads suspended in LiDS lysis buffer were added and incubated for 10 min at room temperature for sufficient mRNA binding. The beads were then transferred to 0.2 ml PCR tubes, washed twice with Wash Buffer A, followed by Wash Buffer B as per the manufacturer's protocol. The beads were finally resuspended in 9 µL of RT-PCR grade DI-water for downstream RT reactions.

UM-UC13 transgene quantification was performed by cDNA synthesis, pre-amplification and targeted qPCR. Synthesis of cDNA was performed directly on oligo-dT beads using the SuperScript III RT Kit (Invitrogen). A 9 µL RNA-bound bead suspension was mixed with 1 µL 10 mM dNTP and incubated at 65 °C for

**Table 1 List of primers used for qPCR.**

| Target | Sequence | Note |
|---|---|---|
| mCherry[a] | CCACCTACAAGGCCAAGAA | Forward |
|  | CGTTCGTACTGTTCCACGAT | Reverse |
|  | TTGGACATC ACCTCCCACAACGAG | Probe |
| EGFP[a] | GCACAAGCTGGAGTACAACTA | Forward |
|  | TGTTGTGGCGGATCTTGAA | Reverse |
|  | AGCAGAAGAACGGCATCAAGGTGA | Probe |
| CD3G | Hs001739941_m1[b] | TaqMan Assay, Thermo Fisher Scientific |
| GAPDH | Hs.PT.39a.22214836 | Integrated DNA Technologies, Inc. |

[a]Custom made primers from Integrated DNA Technologies, Inc., Coralville, Iowa.
[b]_m1 at TaqMan Assay stands for exon span, which implies gDNA is non-detectable.

5 min, followed by chilling on ice for 1 min. The bead suspension was mixed with an equal volume of reaction buffer, consisting of 4 µL of 5× RT buffer, 5 mM $MgCl_2$, 10 mM DTT, 1 µL of RNAseOUT, and 1 µL of SuperScript III RT enzyme was added to the chilled beads (total 20 µL reaction volume) and subjected to thermal cycling (30:00 at 50 °C, 10 cycles of [2:00 at 55 °C, 2:00 at 50 °C], 5:00 at 85 °C). The tubes were held at 4 °C until the next step. Pre-amplification was performed by adding 5 µL of the cDNA-bound bead suspension from the RT reaction to a pool of custom designed probe-based primer assays for EGFP, mCherry, and commercially available GAPDH at final concentration of 0.2× each. Table 1 lists primer sequences used for qPCR. IDT Gene Expression Master Mix was added to a final volume of 20 µL for each isolated sample and 12 cycles of pre-amplification was performed (3:00 at 95 °C, 12 cycles of [0:15 at 95 °C, 1:00 at 60 °C]). The single cell selection was then validated by running qPCR in duplicate, targeting EGFP, mCherry and GAPDH, for each extracted single cell, by using 3 µL of the amplicon from the pre-amplification. To the amplicon, 10 µL of Fast Advance Master Mix (Applied Biosystems), 1× concentration of primer, and qPCR was performed as follows: 2:00 at 50 °C, 2:00 at 95 °C, 45 cycles of [0:01 at 95 °C, 0:20 at 60 °C].

To validate the immunological synapse and prepare samples for sequencing using Jurkat and Raji cells, RNA was eluted from the dT beads, followed by RT reactions using Maxima H minus kit and long-distance PCR for cDNA amplification using Q5 Hotstart Hi-Fi master mix. Specifically, RNA was eluted from oligo dT beads by suspending in 10 µL of elution buffer and incubated at 80 °C for 2 min. The eluted RNA was quickly transferred to a new tube, mixed with 2 µL of 10 µM UMI (unique molecular index) dT primer (IDT), and incubated 2 mins at 85 °C followed by chill in the ice for 1 min. Table 2 lists the oligo sequences used for library preparation. For RT, reverse transcription enzyme Maxima H minus was used according to the manufacture's protocol. Briefly, 4 µL of 5× buffer, 2 µL of dNTP, 1 µL of 50 µM of template switching oligo (IDT) and 1 µL of Maxima H- enzyme mix were added to the 12 µL of RNA samples, attached with UMI. The samples were incubated in a thermocycler (5:00 at 50 °C, 90:00 at 42 °C, 5:00 at 50 °C, 5:00 at 85 °C). The samples were held at 4 °C until the next step. Synthesized cDNA was amplified by long-distance PCR using Q5 Hotstart Hi-Fi master mix according to the manufacture's protocol. For 50 µL reaction, mixture of 25 µL of Q5 master mix and 5 µL of 10 µM handle primer was added to 20 µL of cDNA. After gentle mixing by pipetting, tubes were subjected to thermal cycling (0:30 at 98 °C, 30 cycle of [0:10 at 98 °C, 0:25 at 62 °C, 3:00 at 72 °C], 5:00 at 72 °C). The tubes were held at 4 °C or stored at −20 °C. CD3G and GAPDH transcript quantification was performed by qPCR, under the same conditions.

**PCR clean-up.** PCR amplicons were purified with AMPure XP beads (Beckman Coulter) according to the manufacturer's protocol. Briefly, 35 µL of PCR amplicon were transferred to 96 well plate, mixed with 63 µL of beads and incubated for 5 min at room temperature. Beads were washed with 70% ethanol twice and resuspended in 40 µL of PCR grade water. After incubation for 2 min, the purified-DNAs were eluted by magnet and transferred to a new 96-well PCR plate for library preparation.

**Library preparation and sequencing.** Library was prepared using Illumina DNA flex kit by the Sequencing and Bioinformatics Consortium at the University of British Columbia, following the manufacturer's protocol. In library preparation, we used a custom P5 adapter along with a custom read primer for read1 to read out from the custom site. The prepared library was sequenced using Nextseq 550 with High output, 25 bp READ1(custom) and 125 bp READ2. After sequencing, the UMI was detected using UMI-tools, and then the poly-A tail was removed using Cutadapt. The trimmed fastq file was aligned and mapped using Hisat2 aligner against GRCh38 from NCBI, along with RefSeq annotation. The aligned BAM file was fed into featureCounts in Subread package to count the number of UMI

**Table 2 List of sequences.**

| | |
|---|---|
| UMI-dT sequence | AAGCAGTGGTATCAACGCAGAGTACNNNNNNNNNNNNNNNNTTTTTTTTTTTTTTTTTTTTTTTTTTTTTTTT*(V)*(N) |
| Template switching oligo | AAGCAGTGGTATCAACGCAGAGTGAATrGrGrG |
| Primer for LD-PCR | AAGCAGTGGTATCAACGCAGAGT |
| Custom READ 1 | GCCTGTCCGCGGAAGCAGTGGTATCAACGCAGAGTAC |
| Custom i5 index read primer | GTACTCTGCGTTGATACCACTGCTTCCGCGGACAGGC |
| Custom index-P5 adapter | AATGATACGGCGACCACCGAGATCTACAC [i5] GCCTGTCCGCGGAAGCAGTGGTATCAACGCAGAGT*A*C |
| Index i5 1 [i501] | TAGATCGC |
| Index i5 2 [i502] | CTCTCTAT |
| Index i5 3 [i503] | TATCCTCT |
| Index i5 4 [i504] | AGAGTAGA |
| Index i5 5 [i505] | GTAAGGAG |
| Index i5 6 [i506] | ACTGCATA |
| Index i5 7 [i507] | AAGGAGTA |
| Index i5 8 [i508] | CTAAGCCT |
| Index i5 9 [i509] | CGTCTAAT |

detected per gene. Since featureCounts create BAM tags XT for each gene, and XS for assigned status, we used these tags in count function of UMI-tools to count and deduplicate the reads. Further statistic analyze was performed using edgeR package in R.

**Statistics and reproducibility**. For statistical analyze differential gene expression, the UMI-based count data was loaded to 'edgeR'. The loaded data been grouped first by sample types, and then filtered to remove 0 count genes across the samples. If majority of genes were expressed as 0 count, the data set will be filtered out. After initial filtering, the library normalizing factor was calculated to correct for bias arising from a difference in read counts. The Pearson correlation coefficient within each group was first calculated. The common dispersions were then calculated based on normalized data. Then, the data was fitted using a quasi-likelihood negative binomial generalized log-linear model by the provided function (glmQLFfit). The contrast, which defines group for comparison, was applied and gene-wise statistical tests were conducted with likelihood ratio tests replaced by empirical Bayes quasi-likelihood F-tests (glmQLFtest). The test results were then filtered with FDR corrected, using by $P < 0.05$ as our defined significance threshold. These differential gene expression data was presented by volcano plots, MD plots, or heatmap.

**Reporting summary**. Further information on research design is available in the Nature Research Reporting Summary linked to this article.

## Data availability
Sequence data is available from NCBI BioProject ID PRJNA615050 (Public link). The source data for graphs in the paper is available in Supplementary Data 1.

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

## Acknowledgements

We thank Dr. Peter Black (University of British Columbia) for providing the transgenic UM-UC13 cell lines used in this study. This work was supported by grants from the Canadian Institutes of Health Research (381129, 322375), Natural Sciences and Engineering Research Council of Canada (508392-17, 2015-06541), Prostate Cancer Canada (D2016-1306), University of British Columbia Four Year Fellowship (J.H.L.), Michael Smith Foundation for Health Research (E.S.P. 17963 and J.R.C. 17982), MITACS (J.H.L. IT13817, K.M. IT09621).

## Author contributions

H.M. supervised the study. H.M., J.H.L., and S.P.D. conceived the idea. J.H.L., E.S.P., J.R.C. K.M., and X.D. performed the experimental work. J.H.L. and A.V.L. developed the laser micropatterning system. J.H.L. and S.P.D. analyzed the sequence data. J.H.L., K.M., S.P.D., and H.M. wrote the manuscript.

## Competing interests

H.M. and J.H.L. are listed as inventors on a patent application related to this work. H.M. and J.H.L. are also listed as inventors on a patent application for the cell capture imaging reagent, which has been licensed to MilliporeSigma and is sold commercially as catalogue number LMR001. The remaining authors declare no competing interests.
