## [Peer Review File · Communications Biology]

Reviewers' comments:

Reviewer #1 (Remarks to the Author):

The manuscript by Lee and colleagues reports a novel microwell-based sequencing method, termed See-N-Seq, to link morphological phenotypes to single-cell RNAseq by applying a UV-patterned gelation technique. See-N-Seq enables fast scanning and picking of cells or cell pairs of interest without the loss by physical extraction methods using only existing instrument. The authors investigated the cell-cell conjugates between antigen presenting cells and T cells. Although the sequencing workflow is not truly "single-cell" in nature, the authors were able to find bifurcation towards Th1 and Th2 lineages after 24h of immunological synapse formation of cell pairs. The major advantage of See-N-Seq is to identify and rapidly pick cells with phenotypes that are rare and transient in nature, and thus cannot be inferred from indiscriminate single cell sequencing. The method seems simple and straightforward to implement and may be applied to more biological systems with rare cell-cell interactions or morphological changes. In general, the paper is well written and organized. Some figures and workflow need clarification:

1) Isolation of single cells heavily depends on the spot size of the scanning UV laser. Single cell isolation seems impossible for areas with dense cells. How do authors decide which cells have enough spacing and can be isolated in each well? The time estimate seems too optimistic. Wouldn't the imaging and selection process take over 2.5 hr?

2) The RNA recovery step and library prep seem unnecessarily complicated and laborious. Wouldn't replacing the Dynabeads with barcoded beads, such as Drop-seq beads, improve the throughput? The authors may also increase the throughput by including well indexing oligos, making it a one-pot reaction. Is there any reason that the authors chose a more bulk-like workflow?

3) The text in Fig 2I and 2J are too small. Fig. 2J looks like a barnyard plot, but the axes are percentages, instead of number of unique transcripts. The authors should revise Fig. 2J for clearer representation of the transcript purity. The figure may benefit from adding more data points, instead of just 5 human cells and 5 mouse cells, since the reliability of cell isolation is the key to this method.

4) From Fig 2 and 3, the void still seems large compared with cell sizes. For systems with very rare cell events, can the authors further increase cell density and use tighter beams, such as a 40X objective?

5) In addition to the suspension cells shown here, will See-N-Seq work on adherent cells with large attachment area or more complex contacts, such as neuron synapse or epithelial cells, etc, by using a tighter beam perhaps?

Reviewer #2 (Remarks to the Author):

This report presents a potentially useful and accessible method for isolating single-cells based on imaging, for subsequent RNAseq analysis.

Photo-polymerization conditions (times, power) should be included. It would be helpful if the authors could indicate if these conditions are compatible with commercial systems that may be commonly found in core facilities, including laser scanning confocal microscopes. More details on the custom laser micropatterning system would be appreciated, since currently few details about the components are given.

Is the method compatible with other analytical measurements? For instance, protein or DNA assessment from isolated cells?

It is not quite clear how G1 and G2 were defined, based on S4 does this group consist of only 3 cells? More details should be provided on number of cells analyzed in the different conditions, and identifying statistically meaningful differentially expressed genes is potentially challenging when looking at 3 cells (which were themselves potentially selected based on differential gene expression?).

At what point are the cells fixed or dead in this process, and what is the range of times that the technique will work between the different steps involved? Particularly with respect to the ethanol and non-porous polymer treatment.

Authors may want to reference nanostring technology and compare approaches / tradeoffs.

Other surface anchoring methods are mentioned (e.g., PLK coating) but deemed less appropriate based on potential biological effects. What are the biological effects from UV curing and solvents?

Manuscript ID: COMMSBIO-21-2584-T

Title: See-N-Seq: RNA Sequencing of Target Single Cells Identified by Microscopy

We would like to thank the reviewers for their supportive review. Responses to all their comments and suggestions are provided below.

REVIEWER COMMENTS:

Reviewer #1 (Remarks to the Author):

The manuscript by Lee and colleagues reports a novel microwell-based sequencing method, termed See-N-Seq, to link morphological phenotypes to single-cell RNAseq by applying a UV-patterned gelation technique. See-N-Seq enables fast scanning and picking of cells or cell pairs of interest without the loss by physical extraction methods using only existing instrument. The authors investigated the cell-cell conjugates between antigen presenting cells and T cells. Although the sequencing workflow is not truly “single-cell” in nature, the authors were able to find bifurcation towards Th1 and Th2 lineages after 24h of immunological synapse formation of cell pairs. The major advantage of See-N-Seq is to identify and rapidly pick cells with phenotypes that are rare and transient in nature, and thus cannot be inferred from indiscriminate single cell sequencing. The method seems simple and straightforward to implement and may be applied to more biological systems with rare cell-cell interactions or morphological changes. In general, the paper is well written and organized. Some figures and workflow need clarification:

1) Isolation of single cells heavily depends on the spot size of the scanning UV laser. Single cell isolation seems impossible for areas with dense cells. How do authors decide which cells have enough spacing and can be isolated in each well? The time estimate seems too optimistic. Wouldn't the imaging and selection process take over 2.5 hr?

Regarding cell spacing, our laser spot size is $\sim 1 \mu\text{m}$, but actual patterning spot is enlarged to $12 \mu\text{m}$ due to propagation during curing. Consequently, we need a minimum distance of $12 \mu\text{m}$ between cells in order to adequately select specific single cells. This cell-cell distance is largely the case when we seed 2,000 cells per well. Furthermore, since we select only a single cell per well, we can usually choose between multiple candidates to find a cell that is adequately isolated. We discussed the laser spot size issue and the importance of appropriate cell seeding density on page 7.

Regarding screening time, we believe our time estimate for selecting 384 cells from each well plate in 2.5 hours is appropriate. Imaging cells at 10X required 8 seconds/well, enable the plate to be imaged in 51 minutes. Synapse formation was a sufficiently frequent event that cell selection was performed manually and the time required to select 384 cells was less than 5 minutes. Non-porous hydrogel polymerization took ~ 15 seconds/well and was completed within 90 minutes. More information about timing has been included on page 8.

We also discussed that visual selection of cells is the main bottleneck in this process. Cell selection may take longer when imaging is performed at a higher magnification, or with rare complex phenotypes. Depending on the phenotype, this challenge can be addressed through cell image segmentation and automatic scoring for specific phenotypic features. This issue is discussed on Page 8.

2) The RNA recovery step and library prep seem unnecessarily complicated and laborious. Wouldn't replacing the Dynabeads with barcoded beads, such as Drop-seq beads, improve the throughput? The authors may also increase the throughput by including well indexing oligos, making it a one-pot reaction. Is there any reason that the authors chose a more bulk-like workflow?

We did perform experiments using the barcoded beads from Drop-seq, with the assumption that RNA from a single lysed cell could be captured on an adjacent barcoded bead. However, since the lysate diffusion is not confined (as it is in Drop-seq), we observed contamination over multiple barcoded beads. Furthermore, Drop-seq beads provided much lower RNA yield compared to Dynabeads. We discussed this issue on Page 22.

Introducing a well indexing oligo is a great idea to increase throughput. We are using this approach in the future versions of See-N-Seq.

3) The text in Fig 2I and 2J are too small. Fig. 2J looks like a barnyard plot, but the axes are percentages, instead of number of unique transcripts. The authors should revise Fig. 2J for clearer representation of the transcript purity. The figure may benefit from adding more data points, instead of just 5 human cells and 5 mouse cells, since the reliability of cell isolation is the key to this method.

Thank you for bringing this to our attention. This issue has been fixed in Fig. 2I and 2J.

4) From Fig 2 and 3, the void still seems large compared with cell sizes. For systems with very rare cell events, can the authors further increase cell density and use tighter beams, such as a 40X objective?

We thank the reviewer for this suggestion. The resolution of the See-N-Seq patterning process is currently limited to 12 μm because of polymerization propagation rather than laser spot size. A 40X objective would reduce spot size, but is likely to decrease patterning resolution because the higher numerical aperture increases polymerization propagation. We believe improving patterning resolution will likely require a combination of improvements to the chemical

formulation, as well as optimization of the optical system. We discussed this issue further on Page 7 and in the Discussion on Page 14.

5) In addition to the suspension cells shown here, will See-N-Seq work on adherent cells with large attachment area or more complex contacts, such as neuron synapse or epithelial cells, etc, by using a tighter beam perhaps?

See-N-Seq is currently intended for RNA extraction from non-adherent cells because the current process is limited by its spatial patterning resolution. Using See-N-Seq to isolate RNA from individual adherent cells is more difficult because adherent cells tend to grow adjacent to each other. As a result, using this approach on adherent cells requires significantly better patterning resolution. Nonetheless, we did use See-N-Seq to isolate RNA from adherent cells (NIH-3T3 and UM-UC13) that were enzymatically dissociated. Fig. 2J-K shows the number of mapped UMI arising from RNA extracted from these cells compared to suspension cells. This issue is discussed further on page 10 and 14.

Reviewer #2 (Remarks to the Author):

This report presents a potentially useful and accessible method for isolating single-cells based on imaging, for subsequent RNAseq analysis.

1) Photo-polymerization conditions (times, power) should be included. It would be helpful if the authors could indicate if these conditions are compatible with commercial systems that may be commonly found in core facilities, including laser scanning confocal microscopes. More details on the custom laser micropatterning system would be appreciated, since currently few details about the components are given.

The photo-polymerization conditions including pixel dwell time and laser power have been included to the methods section under *Cell encapsulation and laser micropatterning* on pages 21-22. The details of part number and manufacturer have been added to the methods section under *Laser micropatterning system* on pages 20-21. The components, including the laser, fiberport collimator, galvo-galvo scanner, scan lens, and tube lens, were assembled together. The assembled unit is attached to the rear port of a commercial inverted microscope (Nikon Ti-E) and then adjusted for focusing. A broad range of commercial laser scan systems that attach to the rear port or the camera port of inverted microscopes could be used.

2) Is the method compatible with other analytical measurements? For instance, protein or DNA assessment from isolated cells?

Protein and DNA assessment is possible either through cell imaging or sequencing. Image-based cell selection allows See-N-Seq to take advantage of fluorescence reporters, fluorescence labeling (using antibodies), and image cytometry in order to assess protein expression and localization. Sequencing-based analysis could leverage ongoing advances in multi-omics DNA and protein analysis methods using standard workflow (e.g. CITE-seq) that integrate into scRNA-seq. We have to the discussion the potential to integrate protein and DNA analysis in See-N-Seq in the Discussion section on Page 16.

3) It is not quite clear how G1 and G2 were defined, based on S4 does this group consist of only 3 cells? More details should be provided on number of cells analyzed in the different conditions, and identifying statistically meaningful differentially expressed genes is potentially challenging when looking at 3 cells (which were themselves potentially selected based on differential gene expression?).

G1 and G2 were defined by comparing the correlation of each cell pair based on gene expression. To support the grouping of these cells we have added a dendrogram that uses a distance matrix, derived from the Pearson correlation, to demonstrate that these cells form two distinct clusters.

Further support for the distinction between G1 and G2 is provided by the fact that G1 cells universally over express Tbx21, while G2 cells universally overexpress GATA3. These are regarded as master regulators of the Th1 and Th2 T-cell lineages, and suggest that the distinct clusters constitute different cell subtypes.

This analysis has been clarified on Page 12-13.

4) At what point are the cells fixed or dead in this process, and what is the range of times that the technique will work between the different steps involved? Particularly with respect to the ethanol and non-porous polymer treatment.

Cells are fixed using ethanol early in the process, immediately after imaging the cells in the porous hydrogel. There is some flexibility on when to fix cells because cell viability is retained in the porous hydrogel. However, we typically limit this time to <3 minutes to minimize RNA degradation. After ethanol fixation, non-porous polymer is added to each well and patterned using the scanning laser. Typically, this step is performed within 5 minutes after ethanol fixation. This issue has been discussed on Page 21.

5) Authors may want to reference nanostring technology and compare approaches / tradeoffs.

We thank the review for this suggestion. The Nanostring Digital Spatial profiler (DSP) is a method to associate gene expression analysis with spatial position in cell and tissue samples. The

principle involves in situ hybridization of RNA probes with photocleavable oligo barcodes. DSP offers an opportunity to explore regional gene expression in tissue samples in areas of interest with 50 μm resolution. However, this method has yet to be refined with sufficient spatial resolution for single cell transcriptome analysis.

6) Other surface anchoring methods are mentioned (e.g., PLK coating) but deemed less appropriate based on potential biological effects. What are the biological effects from UV curing and solvents?

Indeed, surface anchoring methods can cause biological alterations because cells are cultured for multiple days on the surface of a slide prior to imaging. In our process, the cells are exposed to pre-polymers only briefly. The pre-polymers are suspended at low density in water and the cells are fixed using ethanol shortly after seeding. Therefore, the biological effects of exposure to these reagents should be minimal because cell culture takes place outside of the microwell plate used for polymerization.

REVIEWERS' COMMENTS:

Reviewer #1 (Remarks to the Author):

The authors have properly and thoroughly answered all my questions and concerns in their letter and manuscript. The authors have made necessary adjustments and greatly improved the figures. I have no other questions regarding the submission.

Reviewer #3 (Remarks to the Author):

Thank you for addressing the comments.